# Mechanisms and Applications of Bacterial Sporulation and Germination in the Intestine

**DOI:** 10.3390/ijms23063405

**Published:** 2022-03-21

**Authors:** Nienke Koopman, Lauren Remijas, Jurgen Seppen, Peter Setlow, Stanley Brul

**Affiliations:** 1Swammerdam Institute for Life Sciences (SILS), University of Amsterdam, 1098 XH Amsterdam, The Netherlands; laurenremijas@gmail.com (L.R.); s.brul@uva.nl (S.B.); 2Tytgat Institute for Liver and Intestinal Research, Amsterdam University Medical Centers, Location AMC, 1105 BK Amsterdam, The Netherlands; j.seppen@amsterdamumc.nl; 3Department of Molecular Biology and Biophysics, UConn Health, 263 Farmington Avenue, Farmington, CT 06030-3305, USA; setlow@uchc.edu

**Keywords:** sporobiota, spore, microbiome, inflammatory bowel disease, probiotics, vaccine vehicle, drug delivery system

## Abstract

Recent studies have suggested a major role for endospore forming bacteria within the gut microbiota, not only as pathogens but also as commensal and beneficial members contributing to gut homeostasis. In this review the sporulation processes, spore properties, and germination processes will be explained within the scope of the human gut. Within the gut, spore-forming bacteria are known to interact with the host’s immune system, both in vegetative cell and spore form. Together with the resistant nature of the spore, these characteristics offer potential for spores’ use as delivery vehicles for therapeutics. In the last part of the review, the therapeutic potential of spores as probiotics, vaccine vehicles, and drug delivery systems will be discussed.

## 1. Introduction

Recent observations, combining observational and omics-based studies, have suggested that microbes inhabiting the human gut and the genes they carry, which make up the expansive and highly diverse human gut microbiome, can have a significant role in gut metabolism and health [1]. Within this large assembly of gut bacteria, archaea, viruses, and fungi, spore-forming bacteria have only recently been indicated as common, commensal occupants [2,3,4]. The sporobiota, defined as the proportion of bacteria capable of endospore formation within the microbiome, may constitute up to 50% of bacteria within the human gut [3,5]. Thus, improved knowledge of the sporulation process, spore properties and germination, transmission dynamics, and consequences of gut colonization will all be important in providing a complete understanding of the gut microbiome and its overall function.

Strategies to isolate and/or identify spore-formers from feces include utilization of various physical and chemical treatments to isolate resistant spores by killing vegetative cells while leaving spores intact. These methods are followed by either culturing with added germinants, such as bile acids and amino acids, or DNA extraction and sequencing [2,6,7,8,9,10,11]. Results include successful culturing of previously “unculturable” species and allowed for identification in the new isolates of a series of genes linked to sporulation [2]. These recent improvements have vastly improved our knowledge of spore-formers and their environments. Previously regarded as primarily soil-dwellers, it is now apparent that both aerobic and anaerobic spore-formers occupy a significant niche within the human gastro-intestinal tract. 

Gut microbes are known to interact closely with the epithelial immune system, priming the differentiation and development of intestinal immune cells while maintaining the important symbiotic relationship between host and microbe. Differences and disruptions of the gut microbiome are implicated in multiple human disorders, but the contribution of spore-formers is relatively understudied. Due to improved isolation and identification procedures, spore-specific interactions can now be studied to better understand the differences between healthy gut microbiomes and gut dysbiosis. 

In this review, we discuss the mechanisms that dictate spore formation, survival, and germination within the gut environment. Some spore characteristics modulate or determine the interaction with the host and therefore show potential to be exploited for therapeutic use in various disorders. Therefore, in the last part of this review we will discuss the potential of spores as therapeutics in the form of probiotics, vaccine vehicles, and drug delivery systems. 

## 2. Sporulation and Germination

### 2.1. Sporulation

Sporulation is the process by which a vegetative cell undergoes a developmental change to form a metabolically inactive and highly resistant endospore. The majority of studies on sporulation have focused on *Bacillus subtilis*, both for its competence and well-studied genome that has led to extensive genetic manipulation, or on *Clostridioides difficile*, because of this organism’s implication in disease and its unique genetic makeup, containing a wide variety of mobile elements, the latter differs from most nonpathogenic spore-formers [12].

It has been suggested that the process of sporulation emerged as a result of gains of two types of genes, the first at the base of the Firmicutes phylum, encoding significant and highly conserved proteins such as Spo0A and RNA polymerase sigma (Sig)-factors and the second at the base of *B. subtilis* and for some members of the Peptostreptococccae family, preceding the *Clostridioides* divison [13]. The second type of genes gained defines species more closely related to *B. subtilis* and is associated with taxonomically restricted genes, while the genes gained associated with the *Clostridiodes* sporulation signature suggest horizontal gene transfer [13]. When comparing *Clostridioides* and Bacilli genomes, the majority of sporulation-specific genes have no orthologues between the two species [13,14]. Indeed, *C. difficile* is more closely related to asporogenous species, such as *Peptostreptococcus anaerobius* and *Eubacterium tenue* [12]. 

Here, we provide a brief overview of the various steps in the sporulation cycle based on what is observed in *B. subtilis*, unless otherwise noted. For a more detailed analysis, turn to the reviews by Errington 2003, Piggot and Hilbert, 2004, Higgins and Dworkin, 2012, and Tan and Ramamurthi, 2015 [15,16,17,18].

#### 2.1.1. Initiation of Sporulation

For most species, sporulation is caused by conditions unfavorable for growth, such as nutrient depletion. Both internal and external signals can signal the vegetative cell to cease growth and form a less metabolically demanding and more resistant state that can survive nutrient poor conditions. Under these conditions, a decrease in guanosine triphosphate (GTP) levels is known to be an internal signal inducing sporulation [19]. Externally, cell density is a factor that has been shown to play a role in the induction of sporulation, as *B. subtilis*, for example, possesses a sporulation-linked quorum-sensing system to detect both nutrient limitation and high cell density [20]. However, it should be noted that certain species, such as *C. difficile*, sporulate continuously at low levels, even in conditions well supporting growth [21].

In *Bacillus* species, sporulation is initiated via a phosphorelay mechanism, controlled by the major protein kinase KinA, as well as other kinases, which induces the phosphorelay by phosporylating Spo0F upon autophosporylation [18,22,23]. Phosphotransferase Spo0B then phosphorylates and activates the master transcription regulator Spo0A, with the help of Spo0F acting as an intermediate phophoryl group receiver upon phosphorylation by multiple sensor histidine kinases. Spo0A directly and indirectly influences the expression of over 500 genes [24,25,26]. However, levels of phosporylated Spo0A can also be decreased by multiple phophatases, and the level of Spo0A~P is an important factor determining whether or not sporulation is initiated. Notably, Clostridia lack phosphotransferases such as Spo0F and Spo0B [27]. Rather, the initiation of sporulation is dependent upon the two RNA polymerase sigma factors SigH and SigA, which drive Spo0A expression and activate putative orphan sensor histidine kinases to directly phosphorylate Spo0A [27,28].

It is also important to appreciate that not all cells in a population sporulate, as sporulation is controlled by a complex set of feed-forward and feedback loops which modulate the number of cells entering sporulation [18]. The process of sporulation is also energetically expensive, becomes irreversible once initiated, and abandons growth, a potential disadvantage in multibacterial natural environments. Therefore, there are many checkpoints upon initiation of sporulation [18]. Regulation of sporulation is tightly controlled by a series of proteins, such as CodY, a well conserved protein expressed in both *B. subtilis* and *C. difficile*, which senses and binds GTP and is capable of repressing sporulation in a nutrient-dependent manner [29,30]. As noted above, phosphatases can counterbalance Spo0A~P production, and the level of Spo0A~P is an important factor in sporulation initiation, although recent findings suggest downstream events also control the commitment to initiate sporulation [18]. All these variable factors ensure that a cell population’s entry into sporulation is heterogeneous, and with some cells never even entering sporulation.

#### 2.1.2. Chromosome Segregation and Cell Division

Upon Spo0A phosphorylation, the vegetative cell contains at least two identical chromosomes, with at least one, and sometimes two, destined to localize in the developing spore, also called the forespore [31]. The chromosomes in the cell early in sporulation form an axial filament, which is attached to the cell poles via interaction with the DNA-binding proteins RacA and DivIVA in *B. subtilis* [32,33]. RacA and DivIVA, in addition to a myriad of other chromosomal organization proteins, ensure segregation of at least one complete chromosome into the forespore [34]. Importantly, in case the cell still contains actively replicating chromosomes or there is significant DNA damage, regulatory protein Sda inhibits KinA-mediated phosphorylation of Spo0F to prevent further sporulation initiation [35,36], while DisA, an additional regulatory DNA-binding protein, can prevent sporulation initiation when the chromosome is damaged [37].

#### 2.1.3. Septation and Differential Gene Expression

Sporulation division is polar, unlike the symmetric division observed in vegetative growth. Asymmetric division, or septation, leads to the formation of two cells, the larger mother cell and the smaller forespore, in which much further gene expression is controlled by compartment-specific synthesis or activation of sporulation-specific RNA polymerase sigma factors. Forespore-localized SigF and mother cell-localized SigE are the primary sigma factors, produced under the control of the Spo0A regulator, although SigF is held in an inactive state by the SpoIIAB protein [18]. Upon completion of septation, SigF is activated in a Spo0A-dependent manner through action of the SpoIIE phosphatase, which results in removal of inhibitory SpoIIAB from SigF in the forespore [15]. Active SigF leads to production of SpoIIR in the forespore, which signals to the mother cell to activate SpoIIGA, eventually leading to SigE activation [15]. The SigE controlled regulon is necessary for the next important step in sporulation, engulfment [15].

#### 2.1.4. Engulfment

Engulfment of the forespore by the mother cell is initiated by proteins produced in the mother cell selectively degrading and rearranging the septum between the mother cell and forespore, leading to the movement of the mother cell plasma membrane around the forespore to form the double-membrane-bound forespore [17]. Engulfment is mediated by a complex of SpoIID, SpoIIM, and SpoIIP proteins located in what will become the outer forespore membrane which directs the mother cell membrane to form around the forespore [38]. Once the forespore has separated from the mother cell membrane and is released into the cytoplasm, further compartment-specific gene expression begins. In particular, prior to engulfment, active SigF in the forespore leads to transcription of SpoIIIG, generating inactive SigG bound to SpoIIAB; upon engulfment, SigG is activated in the forespore, leading to production of the SigG-dependent SpoIVB protein [15,34]. SpoIVB, an intracellular protease, is secreted across the forespore membrane to signal SpoIVB to convert the mother cells’ pro-SigK into active SigK [15]. The action of these various RNA polymerase sigma-factors, in particular SigK in the mother cell and SigG in the forespore, as well as changes in levels of other transcriptional regulators lead to the final transcription changes governing spore formation and subsequent mother cell lysis and spore release [18].

### 2.2. Spore Structure and Survival 

Spores can persist for extremely long times, with some studies suggesting spores can survive for millions of years [39,40]. The structure and chemical composition of the spore account for its resistance against heat, chemicals, and both ionizing (γ) and UV radiation. Here, the different layers of the mature spore and their functions in spore survival will be briefly discussed. Spores of all Firmicute species share a similar structure, although the proteins that constitute the various layers can differ greatly between species and strains resulting in variable survival and germination properties [41].

#### 2.2.1. Exosporium and Crust

The outermost layer of the mature spore, the balloon-like exosporium, is found in only a select number of spore-forming species, including *Bacillus cereus* and *Bacillus anthracis,* as well as *C. difficile* [42]. The exosporium has an outermost nap of filamentous hair-like projections, formed by the glycoprotein *Bacillus* collagen-like protein A (BclA) [42]. The number of amino acid repeats in the BclA protein that determines filament length is variable, and therefore exosporium morphology can be both species- and strain-specific [42]. The exosporium surface also contains enzymes that might affect germination, such as alanine racemase [43]. The hydrophobic nature of the exosporium, specifically of BclA, may also contribute to binding of spores to some surfaces [43]. Interestingly, spore uptake by host phagocytic cells is modulated by BclA, and cells lacking BclA bind more generally to epithelial cells [43]. 

Recently, *B. subtilis* was noted to have an outermost layer named the crust, which mimics the exosporium of other *Bacillus* species in some respects [44]. The crust has been suggested to provide some level of chemical resistance to the spore, but this has yet to be shown directly [45]. It is not known whether the crust functions similarly to an exosporium, or whether proteins in the crust have unique functions. However, some orthologs of *B. subtilis* crust proteins have been identified in the exosporium of other *Bacillus* species [46]. 

#### 2.2.2. Coat

Located below the exosporium is the spore coat, which is made up primarily of protein in a composition that differs greatly between species [47,48]. While spores of *Bacillus* species share ~70% of coat proteins, *C. difficile* spores share less than 25% of spore coat proteins with *B. subtilis* spores [41]. The coat serves as the first line of defense against reactive chemicals including oxidizing agents, as well as enzymes that degrade spore peptidoglycan (PG), the latter by acting as a physical barrier as spore coat proteins are both quite insoluble and highly crosslinked by a variety of covalent bonds [39,47,49,50,51]. In addition, carotenoids in spores’ outer layers can protect spores against UV radiation [52]. 

Beyond providing resistance, the spore coat is also able to sense and respond to external environmental signals, allowed by the coat’s flexibility and species-specific ridges formed during sporulation [53]. Additionally, the coat plays a role in the initiation of germination, as pores in the coat allow low molecular weight germinant signals to reach spores’ more inner layers where germinants are sensed, while preventing the passage of larger molecules, such as lysozymes [50]. A few enzymes are also in the spore coat, including alanine racemase which converts l-alanine to d-alanine [54]. Since l-alanine often acts as a powerful germinant with d-alanine as an inhibitor, coat-bound enzymes could suppress germination until more favorable environmental conditions arise [54,55]. 

In animal models infected with *C. difficile*, it was shown that the major spore coat protein CotE aids in spore adhesion by binding to mucin and further enables spore virulence via mucin degradation [56].

#### 2.2.3. Outer Membrane

Located beneath the spore coat is the outer membrane, which has little known function in resistance or germination, but has been shown to be important in sporulation [16,57]. Due to difficulties isolating the outer membrane from spores, it is unknown whether the membrane is maintained in dormancy [57].

#### 2.2.4. Cortex

The spore cortex is composed of a thick layer of peptidoglycan (PG), which plays an important role in germination, sporulation, and wet heat resistance, the latter most likely by the cortex’s involvement in reducing spore core water content [57,58]. PG in the cortex has a few notable differences from vegetative cell PG, namely the cyclization of many of the *N*-acetylmuramic acid residues into muramic acid-δ-lactam. This modification prevents the cross-linking of many of the cortical PG’s *N*-acetylmuramic acid residues, a characteristic of vegetative cell PG [47,59]. Among various species of spore-formers, the unique cortex PG structure is well conserved [60]. Additionally, in *B. anthracis* and its very close relatives, a combination of *O*-acetylation and N-acetylation of spore cortex PG helps to reduce sensitivity to the innate immune system lysozymes [61]. 

#### 2.2.5. Germ Cell Wall

The germ cell wall, located underneath the spore cortex, is also a PG layer. Unlike the cortex, the structure of the germ cell wall PG is identical to that of vegetative cells [47]. Therefore, levels of cross-linking in this layer are much higher than in the cortex due to the formation of muramic acid-δ-lactam in the cortex. Differences between the chemical structure of the cell wall and cortex PG allow for selective degradation of the cortex during germination, while the germ cell wall becomes the cell wall of the germinated spore [47,60]. 

#### 2.2.6. Inner Membrane

The inner membrane (IM) of the spore, located below the germ cell wall and surrounding the spore core, has a fatty acid and phospholipid composition rather similar to that in growing cells. However, the IM has low permeability, even to water [62,63,64,65,66]. In addition, incorporating a lipid probe into the IM has shown that the IM has a very low lipid mobility compared to the membrane of vegetative cells or germinated spores [67]. Notably, a significant proportion of the dormant spore IM is extruded into the central spore core, but merges with the IM upon spore germination when the core volume expands ~2-fold [68]. In addition, the IM plays a major role in spore germination, as it contains the germinant sensors in spores of most species as well as a channel that is important not only in germination, but also in sporulation, as mentioned below [69]. 

#### 2.2.7. Core

The center of the mature spore, called the core, contains ~25% of its dry weight as CaDPA, a 1:1 chelate of pyridine-2,6-dicarboxylic acid (dipicolinic acid, DPA) and Ca^2+^, and CaDPA accumulation from the mother cell contributes to the core’s low water content, as low as 25% of wet weight [39]. Wet heat resistance, or the ability to survive heat in aqueous solution, is largely determined by the reduced core water content [39]. It has been shown that the higher the spore water content is, the lower the wet heat resistance is [47]. The exact process by which wet heat kills spores is still unknown; this has been shown not to be caused by DNA damage, but wet heat does denature some core proteins which may lead to the death of the spore [39]. Mechanistically, dehydration reduces protein mobility, which could account for the stability of most spore core proteins at high temperatures, as well as the inactivity of core enzymes [39].

The core also contains the spore DNA, which can be as one or two genomes; spore mRNA, ribosomes, and enzymes are also present [31,47]. However, it appears most likely that mRNA in the core is there to provide germinating spores with ribonucleotides for new RNA synthesis, rather than to direct synthesis of new proteins early in spore germination [70]. 

Among the most abundant proteins in the core are a group of α/β-type small acid-soluble spore proteins (SASPs) found only in spores of Gram positive spore forming bacteria and with sequences that are well conserved across species, including *C. difficile* [39,47]. Binding of α/β-type SASPs to spore DNA protects the genetic material against: (i) lethal damage from variety of chemicals including H_2_O_2_, by shielding the DNA from chemical attack and preventing guanine oxidation and cytosine deamination [39,71,72]; and (ii) wet and dry heat, by preventing DNA depurination. In addition, when DNA saturated with α/β-type SASPs is exposed to UV radiation, a unique photoproduct termed the spore photoproduct is formed, instead of the usual UV DNA lesions of cyclobutane-pyrimidine dimers and 6-4 photoproducts [39,73]. Notably, the spore photoproduct is more easily and quickly repaired and in a much more error free process than are cyclobutane-pyrimidine dimers and 6-4 photoproducts, meaning damage caused by UV radiation can be corrected by spores’ DNA repair enzymes in the first few minutes of spore outgrowth and without generating mutations [39,72]. One of these DNA repair enzymes is spore photoproduct lyase, which is found only in spores [74]. While α/β-type SASPs were originally suggested not to play a role in the protection of spore DNA against γ-radiation, more recent studies have suggested that they may provide protection along with the dehydration of the core limiting hydroxy radical generation and thereby slowing production of double strand breaks in DNA in the core [39,75,76]. 

### 2.3. Germination and Outgrowth

Spore germination is the process by which the dormant spore is converted into a vegetative cell. Germination can be divided into three stages: activation, Stage I of germination and Stage II of germination; completion of germination is followed by outgrowth leading to production of a vegetative cell. Notably, activation and Stage I of germination requires no ATP or macromolecular synthesis, and this is most likely also true of Stage II. However, outgrowth requires resumption of metabolism and synthesis of new RNAs and proteins. The following paragraph will address the different germination triggers and the stages of germination. 

#### 2.3.1. Germinants and Germination Receptors in the Gut Environment

Despite being metabolically inactive, spores are capable of sensing the external environment and returning to life during germination. Since germination is an irreversible process, appropriate signals to commit to germination are important to ensure conversion of most spores into growing cells. Within the gut, bile acids and the hosts’ nutritional intake are the major source of germinants for the sporobiota. Germinants derived from food include combinations of sugars, inorganic salts, amino acids, inorganic phosphate, and purine nucleosides [77,78,79].

Most Bacilli and Clostridia species’ genomes contain multiple loci encoding germinant receptor proteins (GRs) located in the spores’ IM which are capable of responding to germinants [80,81,82]. Individual IM GRs are specific for germinants, such as the *B. subtilis* GerA GR, which is specific for l-alanine and inhibited by d-alanine [83]. In at least Bacillales species, all GRs are present in the IM in one or a few large complexes termed germinosomes, which are capable of detecting and responding to various different germinants and in which GRs sometimes work cooperatively [69,82]. However, much less is known about how GRs in the germinosome are activated as well as the subsequent signaling pathway. In some way, activated GRs must transmit signals to downstream germination components, including the SpoVA proteins’ IM channel that takes up CaDPA during sporulation and releases it during Stage I of germination. CaDPA releases then activates the cortex-lytic enzymes that degrade the spore cortex PG in Stage II of germination by recognizing the cortex specific modification, muramic acid-δ-lactam.

Spores are also capable of initiating germination without GR-mediated signals [84]. For example, external CaDPA stimulates germination in a GR-independent manner [84]. This process takes place by CaDPA activation of the cortex lytic enzyme CwIJ, enabling germination to proceed independent of any physiological germinants [84]. However, spores of some Clostridiales species, including *C. difficile,* do not have CwlJ and are thus not germinated by CaDPA. A second germinant that is not recognized by GRs is dodecylamine which germinates spores of all species tested to date, including *C. difficile* spores. This agent opens spores’ SpoVA channels for CaDPA, leading to rapid CaDPA release [84]. One final germinant is high hydrostatic pressure; pressures of 50–300 megaPascals trigger germination by directly activating IM GRs, while even higher pressures, 400–900 mPA, trigger the opening of the CaDPA channel in all spores that have been tested [85,86]. Dodecylamine, which is often used as a surfactant, and high pressure are not directly implicated in spore germination in the gut but are of importance in light of needs for spore eradication in hygiene and food processing. 

Importantly, spores of *C. difficile* and closely related species do not have homologs of IM GRs [87]. Rather, *C. difficile* senses bile acids such as taurocholate and cholate, which are released from the gall bladder into the intestinal environment, via the germination-specific pseudoprotease CspC in spores’ outer layers [82,88,89]. Because the majority of bile acids are reabsorbed in the ileum, *C. difficile* spores’ reliance on bile acids ensures that germination occurs mainly in the small intestine, assuring improved cell viability upon germination [90]. Additionally, *C. difficile* germination is dependent upon co-germinants, such as glycine or calcium, probably to activate CspC [89]. Indeed, calcium-depletion was shown to prevent ileal germination of *C. difficile* spores in a murine model, indicating a biological mechanism that explains why individuals with inefficient calcium absorption are more susceptible to *C. difficile* infections [91]. 

Antimicrobial compounds produced by bacteria can both activate and inhibit germination [78,92]. For example, bryostatin is a known germinant, while staurosporine produced by *Streptomyces staurosporeus* blocks the PrkC kinase receptor needed to sense PG fragments [78]. Importantly, this highlights commensal and competitive behavior between spore-formers and other microbes, specifically applicable in the densely populated environment such as the gut.

A detailed list of spore germinants and the bacterial species in which these are shown to act can be found in Table 1. It is important to note that certain co-germinants are often required in addition to the listed germinants, as well as the fact that different strains respond differently to various concentrations of germinants. For example, *B. cereus*, *B. subtilis*, and *B. anthracis* spores are all capable of germinating solely in the presence of l-alanine, but the addition of different co-germinants can increase the germination rate in a species-specific manner, highlighting an important aspect of bacterial environmental adaptation [93]. Furthermore, for amino acids, the concentrations used in the in vitro studies are often a factor 10 or more higher than concentrations observed in the GI tract [94]. The concentrations of bile acids studied are below the concentration observed in the human small intestine, that has already an average of 10 mM total conjugated bile acids alone [95].

#### 2.3.2. Activation 

The first step in spore germination is activation. Spores of some spore-forming bacteria rely on a short time–high temperature treatment to obtain maximum rates of IM GR dependent germination [119]. The precise mechanism of activation is not known, but it seems to make IM GRs more responsive to germinants [120]. Since spores having undergone heat activation are not committed to germination, activation alone is a reversible process [121]. Understanding heat activation is particularly important when considering food safety, as heat can cause spore activation as opposed to intended bacterial death, resulting in food spoilage and possible infection of the consumer. 

#### 2.3.3. Stage I Germination

In spores of *Bacillus* and some *Clostridium* species, following activation, usually by sublethal heat, germinants pass through spores’ outer layers, with this perhaps facilitated by GerP coat proteins, and then interacting with IM GRs leading to an irreversible commitment to germination. Activated GRs induce the opening of as yet unknown IM ion channels to release large amounts of monocovalent cations including protons from the core, increasing the core pH [122,123]. Proton release is crucial in ultimate activation of phosphoglycerate mutase, an enzyme responsible for using spores’ 3-phosphoglycerate depot to generate ATP soon after germination is complete [122]. Immediately following monovalent cation release, CaDPA is excreted from the core through an IM channel, composed of multiple SpoVA proteins, and CaDPA release is paralleled by water uptake into the core by an unknown pathway [123,124]. This partial rehydration of the core reduces spore resistance to wet heat significantly [84].

In spores of some other Clostridiales species, most notably *C. difficile* and its close relatives, germinants are also sensed by what is termed a GR, but this is not an IM GR, but rather a soluble protein present in spores’ outer layer. This type of activated GR then triggers proteolytic activation of a zymogen of spores’ core lytic enzyme termed proSleC, converting it to the active SleC. This now active enzyme hydrolyses cortex PG, and this triggers CaDPA release, perhaps by activation of the mechanosensitive member of the SpoVA channel proteins by the change in core osmotic pressure due to core water uptake replacing CaDPA. Thus, in these spores’ germination, CaDPA release follows cortex PG hydrolysis, the opposite order from what happens in germination of spores of *Bacillus* species and some *Clostridium* spores as well [28]. 

#### 2.3.4. Stage II Germination

In spores of *Bacillus* species and some *Clostridium* species that are activated by germinants interacting with IM GRs, the second stage of germination is defined by hydrolysis of the PG cortex via the core lytic enzymes CwlJ and SleB, with cortex hydrolysis resulting in full core rehydration and core, IM, and germ cell wall expansion [84,123,125]. Completion of this expansion leads to increased core protein and IM lipid mobility; core hydration is now at 80% of wet weight, which renders the germinated spore vulnerable to wet heat [123]. At this stage, metabolic activities and RNA and protein synthesis begin in order to facilitate the conversion of the germinated spore into a growing cell [125]. SASPs are also degraded to free amino acids at this time, allowing transcription and DNA synthesis, as well as protein synthesis, to resume [84,126]. In addition, enzymes act to repair any DNA damage accumulated during spore dormancy prior to outgrowth [72]. 

Notably, as noted above with spores of *C. difficile* even in some other Clostridial spores that contain IM GRs, cortex PG hydrolysis is also carried out by a core lytic enzyme termed SleC which is present as a zymogen in dormant spores and activated by proteolysis by Csp proteases. It is this event that triggers CaDPA release in these spores, not the direct activation of a germinant sensor protein, but it is not clear how IM GR activation in these *Clostridium* species triggers Csp proteases’ activation. 

#### 2.3.5. Outgrowth

Following degradation of the cortex PG, as well as DNA damage repair, outgrowth can begin followed by normal vegetative growth. Stored energy sources, such as 3-phosphoglycerate as well as amino acids generated from SASPs degradation, are initially utilized for ATP generation and macromolecular synthesis, but exogenous energy sources are needed to complete outgrowth [84,125]. Early synthesized proteins include nucleotide and amino acid biosynthetic enzymes and some transporters [84]. More extensive protein synthesis and then DNA synthesis follows, ultimately leading to vegetative cells.

## 3. The Role of Sporobiota in Gut in Health and Disease

### 3.1. Sporobiota in the Gut, an Evolutionary Trade Off 

As discussed above, sporulation utilizes a large subset of genes and requires significant amounts of energy and time to complete. Kearney et al. showed that lysozyme resistant bacteria are more likely to be shared among multiple individuals, which suggest that these costly resistant states are beneficial for cross-host transmission [10]. In line with this, Browne et al. recently found that loss of the ability to sporulate led to greater relative abundance within an individual host, but lower overall prevalence in the population [127]. Thus, spore-forming bacteria may have lower colonization capacity within a specific niche but are capable of much greater host to host transmission, due to their survival strategy, environmental adaptability, and metabolic capabilities, than more specialized asporogenous species [127]. 

Mother-to-child transmission of spores only appears at a low level, as evidenced by a recent study isolating spore DNA from feces of mothers and children throughout the first two years of life [9]. Colonization of spore-formers was shown to occur within the first years of life, following cessation of breastfeeding, as evidenced by low levels of spore-forming bacteria at birth (0.001%), followed by steady uptake of spore-formers into childhood, as compared with adulthood (up to 50%) [128]. Together, these results suggest that spore forming bacteria are mainly acquired from the environment.

As discussed above, bile acids are germinants, which might indicate that spore forming species have evolved as inhabitants of the intestine by adapting to bile salts for germination in the intestine. Loss of sporulation genes within large taxonomic orders can be traced back to groups such as *Lactobacillales* and *Staphylococcae* [13] and this loss may be linked to bacterial adaptation to nutrient rich environments, allowing for greater growth by discarding expensive sporulation genes [127]. Interestingly, in the study conducted by Browne et al., loss of sporulation was observed least in the gut environment, as compared to soil, the rumen, and human oral sites, affirming that the ability to sporulate and germinate in the intestinal environment is advantageous in fecal–oral transmission [127]. For a more detailed reviews about transmission routes and colonization within infants we refer to Browne et al., 2017 and Egan et al., 2021 [3,129].

### 3.2. Pathogenic Spore Formers

While the majority of spore-forming bacteria do not cause disease in humans, a few species can lead to significant problems in food safety and health. Spores of these species are ubiquitous in nature and commonly found in soil and thus in our food products. The resistant nature of spores presents a serious challenge for disinfection and food processing. Pathogenic spore-forming bacteria mainly belong to the *Bacillus cereus* group, such as *B. cereus*, *B. thuringiensis*, and *B. anthracis*, or to the Clostridia, such as *C. difficile*, *Clostridium tetani*, *C. botulinum*, and *C. perfringens*. Both groups produce toxins that affect tissues ranging from the nervous system to the gut. *Bacillus* species share similar core gene content and high chromosomal synteny but are characterized by the presence of unique, distinguishable plasmids that contribute to differing phenotypes and environmental fitness [130,131]. Additionally, for the pathogenic Clostridia, the toxin-encoding genes are often located on transposable elements or plasmids [132]. Subsequent recombination events have produced a multitude of strains within each species with varying degrees of environmental adaptivity [133,134]. 

*B. anthracis* is the causative agent of the infectious anthrax disease. The anthrax toxin’s lethal factor and edema factor it produces contribute to subsequent incapacitation of immune cells [135]. Further, virulence is enhanced by production of a poly-d-glutamic acid polymer capsule, which aids bacterial survival in macrophages [136]. A recent study suggested that *B. anthracis* spore surface protein BclA mirrors the structure of complement system C1q and interacts with the pulmonary host surfactant protein C, suggesting that the exosporium may play a role in host surface recognition [137]. Further, while BclA mutant spores do not exhibit decreased virulence, they do show decreased cytokine responses in macrophages, indicating a potential immunomodulatory role of the exosporium in reducing innate immune responses for enhanced spore colonization [138]. 

*B. cereus* is a major pathogen of food-borne illness and diarrheal disease, causing considerable concern for the food industry [131]. Emetic disorder occurs as a result of a small peptide cereulide which causes mitochondrial death in insulin-producing beta cells [131]. Diarrheal disease is caused by various pore-forming enterotoxins such as hemolysin BL, non-hemolyic neurotoxin, and Cytotoxin K [131]. These enterotoxins are not heat-stable and are released by vegetative *B. cereus* cells growing within the intestinal tract [131]. Enterotoxins are controlled by the PlcR regulon, a quorum-sensing regulator that activates toxin production in response to cell density [139]. *B. cereus* host adherence is an important aspect of pathogenicity, as enterocyte binding prevents intestinal clearing, allowing time for germination and production of cytotoxic molecules, which are only expressed during vegetative growth [131]. *B. cereus* adherence to epithelial cells is inhibited when FlhA, a flagellar assembly protein, is mutated, suggesting that flagella located on the surface allow adherence to epithelial cells [140].

During sporulation, *B. thuringiensis* produces cytotoxic crystal proteins encoded on plasmids carrying the *cry* gene [141]. Typically considered nonpathogenic in humans, recent studies suggest that the production of certain enterotoxins and hemolysins may cause food-poisoning-like disease in humans [142]. Notably, due to the near identical genomic makeup of *B. cereus* and *B. thuringiensis*, misidentification could account for the current lack of data regarding *B. thuringiensis* associated disease [142]. Similar to *B. cereus*, *B. thuringiensis* species often possess the PlcR regulon linked to enterotoxin production [132,141].

*C. tetani* produces the often-fatal tetanus neurotoxin, leading to paralysis in humans. Vaccine use against inactivated tetanus toxin is clinically safe and effective. Similar to *C. tetani, C. botulinum* produces a paralysis-causing neurotoxin, termed botulinum. Botulinum neurotoxins associate together with hemagglutinin, forming large protease-resistant toxin complexes capable of travelling from the gut to the bloodstream, where these can be transported throughout the body, and ultimately to motor nerve endings, causing paralysis [143]. Adult intestinal botulism, in which *C. botulinum* germinates in the intestine, is rare and occurs only in the absence of gut microbiota [144]. 

*C. perfringens* is a gastrointestinal pathogen associated with intestinal infection, causing a range of symptoms from diarrhea to gas gangrene [145]. Strains can be distinguished by different combinations of various plasmid-encoded toxins, such as beta-toxin, epsilon-toxin, iota-toxin, enterotoxin, and necrotic enteritis B-like toxin, while all strains produce chromosomal encoded α-toxin [145]. Specifically, enterotoxin is associated with food-borne illness and disease in humans [146]. Interestingly, *C. perfringens* is also capable of producing hydrogen sulfide gas, which is associated with gastro-intestinal diseases such as ulcerative colitis [146]. 

Finally, *C. difficile* is the most widely studied pathogenic spore-former and the leading cause of nosocomial antibiotic-associated diarrhea. Changes in the microbial community as a result of antibiotic exposure alter normal primary bile acid metabolism by commensal microbes, which in turn promotes *C. difficile* colonization [147,148]. Spore resistance proves a difficult problem for hospitals and *C. difficile* infections are often linked to widespread hospital outbreaks. The *C. difficile* genome is up to 42% larger than non-pathogenic Clostridia and contains a wide array of mobile elements [149]. One such element, PaLoc, encodes the TcdA and TcdB genes and is regulated by a highly specific regulon, *agr* [149]. PaLoc relies on *agr* quorum-sensing to synchronize cell density information with toxin production [150]. There are significant differences in the *agr* and PaLoc loci between hypervirulent and nonvirulent *C. difficile* strains, emphasizing the importance of these regulons [150]. The toxins TcdA and TcdB bind to and inactivate GTPases which causes actin disassembly and intestinal epithelial cell death [147]. Toxin-induced inflammation is also an important disease characteristic [147]. A third toxin produced by *C. difficile*, known as *C. difficile* transferase, is only associated with a subset of strains, which are linked to more severe disease as a result of toxin-mediated host cytoskeleton disassembly and improved host cell adhesion [147,151]. Pattern recognition receptor activation leads to cascades of inflammatory cytokines, ultimately triggering adaptive immune responses, specifically antibodies targeting *C. difficile* toxins [152]. For detailed information on the *C. difficile* mechanism of infection and immune system activation, we refer to Pechine and Collignon, 2016 and Burke and Lamont, 2014 [152,153].

### 3.3. Beneficial Spore Formers Play a Role in Gut Homeostasis

In contrast with the pathogenic species discussed above, many spore-forming species have been implicated as commensal and are even central to a healthy gut microbiome. Commensal spore-formers include members of the Clostridium clusters XI and XIVa, such as *Clostridium leptum*, *Clostridium scindens,* and *C. innocuum* [2,3,132]. Further, species from the *Ruminococcaceae* family, such as *Flavonifractor plautii* and *Ruminococcus bromii* were recently implicated as spore-forming commensals of the gut [2,154]. Other common intestinal species, such as *Eubacterium rectale* and *Eubacterium elegans* may also be capable of spore-formation [2]. Specifically, short chain fatty acids (SCFAs) generated by spore forming bacteria that break down dietary fibers, are linked to improved gut homeostasis, suppressed inflammatory responses, and epithelial barrier functioning [155,156,157,158].

While the metabolic interactions between the host and these bacteria are elucidated increasingly, the interactions between these species while in their spore form remain understudied. However, recognition of spores by the host immune system is observed. *B. subtilis* and *B. anthracis* spores were shown to be sufficient to promote development of gut-associated lymphoid tissue (GALT) [159,160]. Development of GALT, comprising the Peyer’s patches and lamina propria, is integral in healthy gut immunity and controls the expansion and maturation of B cells, which in turn secrete IgA to promote bacterial tolerance and prevention of bacterial leakage across the intestinal epithelium [161]. The stimulation of B cells by spores leading to GALT development is suggested to be driven by a superantigen-like mechanism, in which the spore surface acts as a special antigen to poly-clonally drive B cell activation, promoting GALT development [160]. Thus, commensal bacterial spore-formers can play an important role in immune development.

Beyond GALT development, *B. subtilis* spores also contribute to B and T lymphocyte proliferation within the Peyer’s patches [162]. In addition, *B. subtilis* spores potentially prevent enterotoxigenic *E. coli* infection, which causes traveler’s diarrhea, as was shown by infecting mice with pathogenic *Citrobacter rodentium*, leading to reduced colonization, minimized epithelial damage, and maintenance of T lymphocytes [163,164]. In some human populations *B. subtilis* excludes *Staphylococcus aureus* colonization of the intestine, likely by inhibiting *Staphylococcus aureus* quorum sensing [165]. T lymphocytes play a pivotal role in distinguishing between commensal and pathogenic bacteria; dysregulation of the balance between the different T cells, respectively, Tregs and T helper (Th), can lead to autoimmunity and aberrant immune responses against commensal microbes [166,167,168,169]. 

The interaction between the host and the microbiota works in both directions, the microbiota not only excrete metabolites that affect the host, but the gut environment, determined by both microbiota in the host, also influences which species can colonize. For example, spore-formers, in particular, have been implicated in impacting gut serotonin levels by affecting the host production levels [170], while increased gut serotonin levels, by oral supplementation or inhibiting the serotonin reuptake receptor, led to a higher abundance of spore-formers [171]. Specifically, *Turibacter sanguinis*, a Gram-positive spore-former, was capable of sensing and inducing intestinal serotonin production, promoting its competitiveness for colonization within the host [171]. In another example, butyrate can prevent *C. difficile* infection by stabilizing the hypoxia-inducible factor 1 α, a transcription factor linked to cellular junction and barrier integrity, effectively limiting inflammation and protecting the epithelium against *C. difficile* toxins [155,172]. 

An overview of the interactions of pathogenic and commensal spore formers with the intestine is shown in Figure 1.

## 4. Potential of Sporobiota as Treatment

Various human disorders are associated with dysbiosis and depletion of beneficial species, therefore, modification of the gut microbiota seems to be an attractive solution. Spores represent an appealing vehicle for therapeutic use, due in part to their highly resistant makeup, ensuring stable shelf lives and increasing the likelihood of survival within the gastro-intestinal tract [164]. Below, we will discuss some of the modern uses of bacterial spores as well as currently developing strategies.

### 4.1. Spores as Probiotics

Probiotics have been studied extensively for many years and have proven helpful in the management of disorders involving inflammation such as inflammatory bowel disease and metabolic syndrome [173]. Spore-forming bacteria constitute perfect candidates for probiotic use, as they can be safely processed and purified in a laboratory setting, ensuring delivery of specified species, including strict anaerobes. Finally, the use of spores as probiotics is far less invasive than other microbiota restoring treatment options, such as fecal microbiota transplantation. 

Bacterial spores, specifically those of the *Bacillus* genus, have been used as probiotics in humans since the 1960s to treat various disorders linked to the gastro-intestinal tract. *B. subtilis* spores of the variety Natto are consumed in Japan, as they are the main microbial species in the fermented soybean product Natto. The consumption of Natto has been linked to immune stimulation, lymphocyte proliferation, and beneficial changes in microbiome constitution [174]. 

The potential for spore treatment in abrogating gastro-intestinal inflammation has also been shown in animal models. Administration of a blend of five different *Bacillus* species’ spores, known as MegaSporeBiotic, in rats with induced colitis resulted in reduced inflammation and colonic tissue damage [175]. Specifically, production of inflammatory cytokines TNF-α and IL-6 was markedly decreased [175]. Additionally, levels of ICAM and VCAM-1, which contribute to gastrointestinal inflammation, were reduced [175]. In line with this, a study in mice with induced colitis showed that treatment with *Bacillus coagulans* spores in combination with green banana starch led to an increase in butyrate production, as well as increased levels of the anti-inflammatory cytokine IL-10, ultimately diminishing disease [176]. 

In humans, *Bacillus* spores have been shown to be beneficial in gastro-intestinal disorders. A recent study utilized *B. subtilis* and *B. coagulans* spores to treat adults with functional dyspepsia [177]. Patients given spore probiotics, as compared to placebo, more often reached clinical endpoint [177]. Administration of *B. subtilis* spores has also been shown to be as beneficial in Irritable Bowel Syndrome patients as a low FODMAP diet or antibiotics [178]. Similarly, *Bacillus clausii* spores were recently found to be an effective probiotic treatment for children with acute diarrhea [164,179]. Further, probiotic supplementation with *B. coagulans* in the elderly was shown to increase levels of butyrate-producing, spore-formers such as *Faecalibacterium prausnitzii*, *Clostridium lituseburense*, and *Bacillus* spp., suggesting a symbiotic relationship [180]. A placebo controlled study showed that *B. subtilis* spores improved Bristol stool scale and stool frequency in volunteers with loose stools [181]. 

Fewer studies have been performed with spores from other Firmicutes including *Clostridium* species. One study showed that a probiotic mixture of a mixture of Firmicutes spores isolated from healthy donor stool was effective in recurrent *C. difficile* infected patients as 86.7% of patients reached disease end-point, suggesting a viable and non-invasive treatment for chronic *C. difficile* infection [182]. In ulcerative colitis, supplementation with Firmicutes spores isolated from donor stool was also shown to be effective, although priming with antibiotics led to higher remission rates in patients [183]. This suggests that effective outgrowth of spores and subsequently colonization may need to be induced by first disrupting the intestinal microbiota. Therefore, providing a favorable environment for germination and colonization is an important consideration in designing spore-based treatment. Prebiotic treatment with, for example, fiber can promote outgrowth of the spores by providing necessary nutrients while enhancing production of beneficial SCFAs [156].

### 4.2. Spores as Vaccine or Drug Vehicles 

While wild-type spores offer potential as therapeutics, spores can also be genetically manipulated to carry immunogenic peptides; thus, acting as vaccine vehicles. The temperature stable nature of spores offers a shelf-stable and therefore economically sound alternative to vaccine vectors such as attenuated viruses or RNA, which often need to be stored at freezing temperatures. Commensal spore-forming bacteria might be a safe alternative to viral vaccine vectors, which have the rare potential to convert or cause unwanted hyperimmune reactions [184,185]. On top of that, oral delivery of spores eliminates the need for sterile handling and administration by medical trained personnel. Finally, oral delivery and subsequent spore interaction with mucosal linings can generate both mucosal and humoral immune responses, potentially increasing the efficacy of spore vaccines [185].

Endospore vaccine vehicle production is accomplished via a few different strategies. The general system relies on anchor proteins, such as the common *Bacillus* spore coat proteins CotB and CotC, bound to the antigen of interest. While the majority of successful spore vaccines have displayed antigen on the outer coat, fusion to OxdD, an inner coat protein, could protect vulnerable proteins [186,187]. Recombinant methods utilize preconstructed vectors expressing the target antigen fused to a spore coat protein [188]. Linker peptides can be included to limit rigidity between the antigen and fusion protein [186]. Non-recombinant fusion solutions depend on passive chemical attractions, such as hydrophobic or electrostatic interactions, or the use of cross-linkers, such as glutaraldehyde, to bind desired proteins to the spore surface [186]. Currently, only single antigens have been integrated with the spore surface, but the potential exists to present multiple proteins on a single spore.

A compiled list of spore vaccines developed to target human diseases and the animal models in which these have been tested is shown in Table 2. These recent animal studies utilizing spore vaccine vehicles have shown elicited immune responses including the increase in IgG and IgA production, expression of antigen-specific antibodies and T-cell responses [189]. Interestingly, in one study the gut microbiota composition was examined after oral and intraperitoneal delivery of *B. subtilis* spores engineered to express paramyosin of *Clonorchis sinensis* in mice and no difference in the abundance or diversity between immunized and non-immunized mice was found, suggesting that recombinant spore vaccines do not significantly alter the microbiota when utilized as vaccines [190]. 

Spore vaccine production without genetically modifying the spore producer is also possible by non-covalent binding of toxin antigens to *B. subtilis* spores, although the efficacy of these spores as vaccines has not yet been tested [191]. However, studies have suggested that antigens bound to spores are more stable than their free counterparts [192,193]. Recently, mice nasally immunized with BclA2, a *C. difficile* antigen, non-recombinantly fused to *B. subtilis* spores, showed immune-specific responses, suggesting that non-recombinant fusion is an effective alternative approach [194].

Similar to vaccine vehicles, spores can also be used to deliver drugs of interest to the gut lumen. The techniques to achieve this mirror those used for vaccine development. Enzymes, such as cellobiose 2-epimerase and perocideroxin, have been successfully bound to the spore surface in non-recombinant manners [184]. An anti-inflammatory enzyme bromelain fused to *B. subtilis* spores was also shown to exhibit a much greater stability than the free enzyme [184,217]. Further, animal studies with human proinsulin fused to recombinant *B. subtilis* spores and recombinant *B. subtilis* spores carrying glucagon-like peptide 1 showed promise for treatment of type 2 diabetes [218,219]. 

Spores can also be genetically engineered to carry genes coding for proteins with a therapeutic purpose, which are synthesized upon spore germination. Specifically, use of spore-formers in cancer treatment is being studied. The spore-forming species *Clostridium beijerinckii* was recombinantly engineered to carry *E. coli* nitroreductase, an anticancer enzyme [220]. This species was chosen for its selective ability to grow in hypoxic environments, which are indicative of tumor areas. The study found that spore outgrowth was indeed highly specific for tumors, suggesting a novel method to selectively deliver cancer therapeutics [220]. This method allows for mucosal or intravenous delivery of spores, as opposed to other treatments which rely on direct access to tumor sites. Currently, spores of *Clostridium novyi* are used in clinical trials for treatment of solid tumors as well [221]. A further study used *B. subtilis* spores to deliver curcumin, a targeted colon cancer drug, upon germination in the colon. Results indicated that drug delivery was specific to the colon and led to decreases in cancer cells [222]. 

A new application of spore-formers is in the production of nanobodies, molecules derived from single chain variable antibody fragments of camelids that have great stability and are specialized to target cells and deliver drugs or present antigens. Previous research has utilized *E. coli* to produce nanobodies, but spore-formers offer production advantages due to spores’ stability and resistance [223]. A recent study indicated that *B. subtilis* engineered to secrete nanobodies carrying either enzymes promoting synthesis of small molecules such as caffeine or antigens such as cytotoxic-T-lymphocyte-associated protein 4, successfully produced the desired molecules at high yields, while still being able to form viable and highly resistant spores [223].

## 5. Challenges in the Use of Spores as Therapeutics

Several knowledge gaps and challenges persist in our understanding of spore-forming bacteria and their potential use in therapeutics. One major limitation preventing further research on spores in the gut is the ability to accurately isolate and sequence the spore-forming population in the gut. While advances have been made utilizing spore resistance properties to select for spores, it is known that spores have variable differences in resistance properties [224]. It is important to note that methods such as ethanol isolation favor sporulated species, but total elimination of ethanol-resistant, non-spore-formers is not feasible. Additionally, germination of isolated spores is also complicated by differences in germination features such as GRs and co-germinant requirements between spores of different species and strains. When isolated and in culture, it can be a challenge to produce sufficient numbers of spores since the sporulation characteristics and conditions of many gut spore-formers are still unknown. 

While delivery of spore vaccines has proven effective in animal models, clinical trials in humans have not yet been conducted. The effects of diet, gut motility, and transit time may affect the efficacy of spores as vaccine or drug delivery vehicles by altering spores’ abilities to bind to the gut epithelium and evoke immune responses. This binding is already limited by low binding properties and low protein expression levels in the spore. However, workarounds such as systems utilizing cellulosome-derived cohesin–dockerin interactions to increase the binding affinity of the target proteins to spore surface anchors, have improved efficiency [225]. Recent developments in spore engineering have suggested genetic alteration of the spore coat proteins to induce additional bacterial adhesion molecules, encouraging stronger epithelial binding [226]. 

An additional issue regarding recombinant spore use is the potential for recombinant, resistant spores to exist indefinitely in the environment [227]. One method to prevent this is construction of recombinant spores with disrupted thymidylate synthase genes, generating spores which upon vegetative growth are unable to survive due to minimal thymine synthesis [227]. Another method, termed “germinate to eradicate”, relies on germinants to trigger spore germination into more vulnerable vegetative cells, which are more easily killed by heat or chemicals [228]. Nutrient germinants, such as l-alanine, can be used to specifically target spores of certain species [228]. Despite advances in eliminating spores, evolution of new, more resistant strains poses the potential for more serious food-borne illness outbreaks, demonstrating the need for continued development of spore killing strategies [228].

## 6. Conclusions

The definition and composition of a healthy gut microbiome is still disputed and the role of spore formers in the gut microbiota has only recently started to be established, as isolation and sequencing methods have improved. Species previously thought to be non-spore forming, such as those within the *Ruminococcus* genus, have been identified as possessing sporulation genes, suggesting many common gut inhabitants may actually be capable of sporulation [2].

While certain sporulating bacteria are known pathogens, linked to foodborne illness, toxicity, and intestinal disorders, the establishment of the role of commensal spore-formers as commensal and beneficial members of the gut microbiome affecting gut homeostasis is in progress. The stable and resistant nature of spores and the possibility to germinate and grow in a gut environment, makes them suitable to use as treatment in the form of probiotics, and vehicles for vaccine and drug delivery. Spore-based treatments have shown great promise in animal studies, but human trials need to be advanced further. Nonetheless, spores might open the door to the development of safe, effective, and easy to administer therapeutics. 

## Figures and Tables

**Figure 1 ijms-23-03405-f001:**
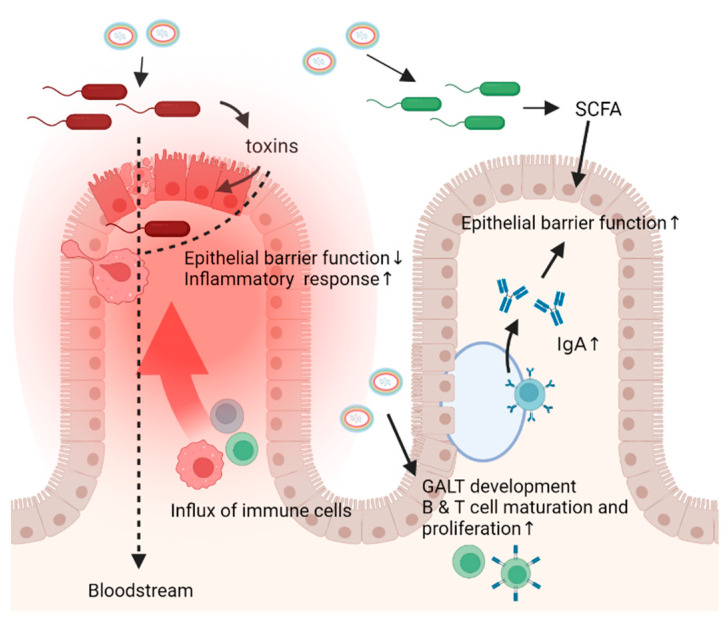
Interactions between spore forming bacteria and the intestine. Spore forming bacteria can act in the intestine both as spore and vegetative cells after germinating under the influence of germinants such as bile acids, amino acids, and other compounds. On the left, interactions with pathogenic spore forming bacteria are illustrated. Once sporulated, these bacteria produce toxins that act locally by interacting with the epithelium and altering immune responses, leading to cell death, tissue damage, and subsequently reduced epithelial barrier function causing a leaky gut. When this happens, inflammatory responses are evoked, and certain bacteria and toxins can travel from the gut to the bloodstream leading to damage in other places in the body such as motor nerve endings which ultimately leads to paralysis. On the right, interactions with beneficial spore forming bacteria are illustrated. Spores can induce GALT development and maturation and proliferation of T and B cells. Mature B cells produce IgA which benefits the epithelial barrier function and promotes bacterial tolerance. This epithelial barrier function is also improved by the action of SCFA, produced by vegetative commensals which are often spore formers. In addition, SCFA are linked to improved gut homeostasis and suppressed inflammatory responses.

**Table 1 ijms-23-03405-t001:** An overview of various spore germinants and the protein triggering germination in different species.

Germinant	Activated Protein	Bacteria	Reference
**Sugars**	GR		
d-glucose		*Bacillus megaterium*	[96,97]
d-mannose		*Bacillus megaterium*	[96]
2-deoxy-d-glucose		*Bacillus megaterium*	[96,98]
d-glucosamine		*Bacillus megaterium*	[96,98]
*N*-acetyl-d-glucosamine		*Bacillus megaterium*	[98]
Cellobiose		*Bacillus megaterium*	[98]
Dextrose		*Bacillus megaterium*	[98]
Maltose		*Bacillus megaterium*	[98]
Methyl-alpha-d-glucoside		*Bacillus megaterium*	[98]
Sorbose		*Bacillus megaterium*	[98]
Starch		*Bacillus megaterium*	[98]
Xylose		*Bacillus megaterium*	[98]
**Purine nucleosides**	GR		
Inosine		*Bacillus cereus*	[99,100]
		*Bacillus anthracis*	[101]
**Amino Acids**	GR		
l-alanine		*Bacillus subtilis*	[99,100]
		*Bacillus anthracis*	[101]
		*Bacillus cereus*	[99,101]
		*Bacillus licheniformis*	[102,103]
		*Bacillus thuringiensis*	[104]
		*Clostridium botulinum*	[105,106]
		*Clostridium sordellii*	[107]
		*Clostridium sporogenes*	[105]
l-valine		*Bacillus subtilis*	[108]
		*Bacillus licheniformis*	[103]
l-asparagine		*Bacillus subtilis*	[109]
		*Clostridium perfringens*	[110]
l-proline		*Bacillus megaterium*	[97]
l-cysteine		*Clostridium perfringens*	[110]
		*Clostridium botulinum*	[111]
		*Bacillus licheniformis*	[103]
l-threonine		*Clostridium perfringens*	[110]
l-serine		*Clostridium perfringens*	[110]
		*Clostridium botulinum*	[105]
l-asparagine		*Clostridium perfringens*	[110]
l-methionine		*Clostridium botulinum*	^[105]^
l-phenylalanine		*Clostridium botulinum*	^[105]^
		*Clostridium sordellii*	[107,112]
l-arginine		*Clostridium sordellii*	[112]
Glycine		*Clostridium botulinum*	^[105]^
l-glutamine		*Clostridium perfringens*	[110]
**CaDPA**	CwIJ		
		*Bacillus megaterium*	[113]
		*Bacillus subtilis*	[114]
		*Clostridium sporogenes*	[105]
**Dodecyclamine**	SpoVA channel		
		*Bacillus thuringiensis*	[104]
		*Bacillus megaterium*	[115]
		*Bacillus subtilis*	[114,116]
**Peptidoglycan**	Protein Kinase		
		*Bacillus subtilis*	[60,78]
**Pressure**			
Low Pressure (100–350 mPa)	GR	*Bacillus subtilis*	[85,86]
		*Bacillus cereus*	[117]
High Pressure (500–1000 mPa)	SpoVA channel	*Bacillus subtilis*	[85,86]
**Bile salts**	Csp		
Taurocholate		*Clostridioides difficile*	[2]
		*Clostridium innocuum*	[2]
		*Clostrdium hathewayi*	[2]
		*Flavinofractor plautii*	[2]
		*Clostridium baratii*	[2]
		*Clostridium thermocellum*	[2]
		*Clostridiaceae*	[11]
Cholate		*Clostridioides difficile*	[2]
		*Clostridium innocuum*	[2]
Glycocholate		*Clostridioides difficile*	[2]
		*Clostridium innocuum*	[2]
		*Lachnospiraceae*	[11]
		*Clostridiaceae*	[11]
Taurochenodeoxycholate		*Clostridiaceae*	[11]
Glycochenodeoxycholic acid		*Clostridioides difficile*	[118]
		*Clostridiaceae*	[11]
Glycodeoxycholic acid		*Clostridioides difficile*	[118]
		*Clostridiaceae*	[11]
Salts	GR		
KBr		*Bacillus megaterium*	[97]

**Table 2 ijms-23-03405-t002:** An overview of developed vaccine strategies against human pathogens utilizing *B. subtilis* spores. The target protein, the disease/pathogen it belongs to, the vector used, the carrier protein, and the animal model used to study the effects in are listed.

Target Protein	Disease/Pathogen	Vector	Carrier	Model	Reference
TTFC	*Clostridium tetani*	pGEM	CotB	Mouse	[193]
PA	*Bacillus anthracis*	pDG364	CotB/CotC	Mouse	[195]
GST-Cpa247-370	Necrotic enteritis	pDG1664	CotB	Mouse	[196]
Toxin A/B	*Clostridioides difficile*	-	CotB	Mouse	[197]
FliD	*Clostridioides difficile*	pDL	CotB	-	[198]
VP1	Enterovirus 71	pDG1662	CotB	Mouse	[199]
M2e3	Influenza virus	pDG1664	CotB	Mouse	[200]
MPT64	*Mycobacterium tuberculosis*	pcotVac	CotB	Mouse	[201]
TTFC/LFB	*Clostridium tetani/E. coli*	pRH22/pIM51	CotC	Mouse	[202]
CsTP22.3	*Clonorchis sinensis*	pGEX	CotC	Rat	[203]
SjGST	Schistosomias	pGEX	CotC	Mouse	[204]
UreB	*Helicobacter pylori*	pUS186	CotC	Mouse	[205]
Enolase	*Clonorchis sinensis*	PEB03	CotC	Rat	[206]
CsPmy	*Clonorchis sinensis*	PEB03	CotC	Mice	[190]
CagA	*Helicobacter pylori*	-	CgeA	-	[188]
TTFC	*Clostridium tetani*	pET28b	CotB	Mouse	[195]
TcdA	*Clostridioides difficile*	pET28b	CotB/CotC	Mouse	[207]
OmpC	*Salmonella*-serovar Pullorum/Typhimurium	pDG364	CotC	Mouse	[208]
CsCP	*Clonorchis sinensis*	pEB03	CotC	Mouse	[209]
TP20.8	*Clonorchis sinensis*	pGEX	CotC	Rat	[210]
CsLAP2	*Clonorchis sinensis*	PEB03	CotC	Mouse	[211]
UreA	*Helicobacter pylori*	pGEM	CotC	-	[212]
SEB	*Staphyloccocus aureus*	pET28a	CotC	Mouse	[213]
gC/gD	Pseudorabies virus	p43NMK	-	Mouse	[214]
CsSerpin3	*Clonorchis sinensis*	PEB03	CotC	Mouse	[215]
PA	*Bacillus anthracis*	pMar3g	-	Mouse	[216]

## Data Availability

Not applicable.

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
