# Peer review of "Mechanisms and Applications of Bacterial Sporulation and Germination in the Intestine"

_ijms, 2022, doi:10.3390/ijms23063405_

Round 1

Reviewer 1 Report

The manuscript reviews the sporulation and germination processes and spore properties and the potential application of spore-forming bacteria to be used as probiotics and delivery vehicles for therapeutics. The manuscript is well written, structured and presents relevant and updated information on the topic. Then it deserves publication, almost as it is. Only some minor remarks:

1.- Please define GTP and SCFAs at first use (line 75 and 422, respectively).

2.- Stages of germination are referred as I and II or 1 and 2 in different parts of the manuscript. Please, use only one of them.

3.- Bacillus and Clostridium genus are typed without italics in some places (lines 291, 305, 308 and 514, among others).

4.- Line 599: Heading 4.5 should be 5.

Reviewer 2 Report

The review entitled: "Mechanisms and applications of bacterial sporulation and germination in the intestine" is very interesting and it is a promising field of research that will deserve further attention and clinical studies.

However, to demonstrate how sporobiota is contributing to gut homeostasis should be better studied, compiled and discussed. Authors should done an effort for focusing on the main scope of the review.

The theoretical approach shared in this review regarding sporulation and germination are valuable, however the link with human gut interaction is not well supported and the clauses are very general (from page 2-page 7 are general info).

Only the section 2.3.1. Germinants and germination receptors in the gut environment it can be seen the plausible interactions and linked mechanisms.

Therefore, I would suggest to summarise previous sections and do it more readable (table, figure, references, etc).

Please revise also the section 4.2. to 4.5 (4.3 and 4.4 do not exist). 

Fig. 1 It is not giving a sound supported view on the differential impact of pathogen/commensal spores:

Please revise it and also the text connected to this Figure.

4.4. and 4.5 can be part of the same section as both describe spores as treatments/therapeutics

Moreover, I would suggest to develop the spores as probiotics and main related point to gut homeostasis.

Reviewer 3 Report

The main topic of this review was to give an overview of spore formation and their germination as well as the therapeutic potential of spores as probiotics, vaccine vehicles and drug delivery systems. Different parts of the review largely differ in quality of presented information. The most weaknesses of the review are described in major points. There are additional suggestions for changes and typos pointed out in minor points.

Major point:

  1. Parts of the review will require additional work and rewriting. Among these parts belongs description Initiation of sporulation, Chromosome segregation and cell division, Septation & differential gene expression and Engulfment. These parts are too incomplete, vague and inaccurate. Below are some of these critics.
  2. Line 73: What is the meaning “sporulation is often caused by unfavorable environmental conditions…”? Sometimes is then caused by favorable conditions?
  3. Line 76: “Other external factors, such as cell density, have also been implicated as playing a role…” Many of these external factors have been well characterized and not just implicated.
  4. Line 80: The phosphorelay mechanism is not controlled only by the major protein kinase KinA and other kinases are well described each of them signaling different environmental and cell distress.
  5. Line 82: It is written that Spo0F and Spo0B phosphorylate Spo0A. It is not true because only Spo0B has that function and Spo0F is only intermediate phosphor group receiver.
  6. Line 83: “Spo0A directly influences the expression of over 500 genes” that is not true. It does not bind to promoters of over 500 genes.
  7. Line 84: “the initiation of sporulation is dependent upon the RNA polymerase sigma factor SigH” this is not true also SigA is required.
  8. Line 87: “It is also important to appreciate that the decision to sporulate is not a homogenous cell population decision.” Why is this important to appreciate?
  9. Line 89: “The process of sporulation is also energetically expensive and irreversible once begun, so cell
    populations are careful to commit to it.” This statement is not true.
  10. Line 98: “with at least one destined to localize in the developing spore, also called the forespore.” That means that 2 or more chromosomes can be localized in the forespore?
  11. Line 102: “ensure segregation of at least one complete chromosome into the forespore…” same as above point 10.
  12. Line 120: “Once the forespore has separated from the mother cell membrane and is released into the cytoplasm, compartment-specific gene expression begins.” Compartment specific gene expression begins with activation of SigF in the forespore.
  13. Line 122: “generating inactive SigG bound to SpoIIAB” Why SpoIIB is not mentioned above as the anti sigma factor of SigF?
  14. Line 131: “Spores of all Gram-positive bacteria share a similar structure, although the proteins” Spores of all G+ bacteria? How about for example Streptomyces’ spores?
  15. Line 201: “The core also contains the spore DNA, which can be as one or two genomes; spore mRNA, ribosomes, and enzymes are also present [44].” Are you sure about two genomes? Reference [44] is very old from year 1970 and this reference does not mention two genomes.
  16. The whole part of sporulation process should be rewritten in more rigorous way.
  17. Some texts under subtitles are very short just 2 sentences and I would suggest restructuring these parts of the manuscript, either by elaborating in these specific topics or combining some parts.

Minor points:

  1. Check for all typos.
  2. SigF should be SigF or sF and this should be corrected for all sigma factor names in the manuscript.

Round 2

Reviewer 3 Report

Dear authors,

the revised version of the manuscript is improved substantially.